# Computational Characterization of Membrane Proteins as Anticancer Targets: Current Challenges and Opportunities

**DOI:** 10.3390/ijms25073698

**Published:** 2024-03-26

**Authors:** Marina Gorostiola González, Pepijn R. J. Rakers, Willem Jespers, Adriaan P. IJzerman, Laura H. Heitman, Gerard J. P. van Westen

**Affiliations:** 1Leiden Academic Centre of Drug Research, Leiden University, Einsteinweg 55, 2333 CC Leiden, The Netherlands; m.gorostiola.gonzalez@lacdr.leidenuniv.nl (M.G.G.); pepijnrakers@hotmail.com (P.R.J.R.); w.jespers@lacdr.leidenuniv.nl (W.J.); ijzerman@lacdr.leidenuniv.nl (A.P.I.); l.h.heitman@lacdr.leidenuniv.nl (L.H.H.); 2Oncode Institute, 2333 CC Leiden, The Netherlands

**Keywords:** computational drug discovery, membrane protein, cancer, anticancer target, receptor tyrosine kinase, RTK, G protein-coupled receptor, GPCR, solute carrier, SLC

## Abstract

Cancer remains a leading cause of mortality worldwide and calls for novel therapeutic targets. Membrane proteins are key players in various cancer types but present unique challenges compared to soluble proteins. The advent of computational drug discovery tools offers a promising approach to address these challenges, allowing for the prioritization of “wet-lab” experiments. In this review, we explore the applications of computational approaches in membrane protein oncological characterization, particularly focusing on three prominent membrane protein families: receptor tyrosine kinases (RTKs), G protein-coupled receptors (GPCRs), and solute carrier proteins (SLCs). We chose these families due to their varying levels of understanding and research data availability, which leads to distinct challenges and opportunities for computational analysis. We discuss the utilization of multi-omics data, machine learning, and structure-based methods to investigate aberrant protein functionalities associated with cancer progression within each family. Moreover, we highlight the importance of considering the broader cellular context and, in particular, cross-talk between proteins. Despite existing challenges, computational tools hold promise in dissecting membrane protein dysregulation in cancer. With advancing computational capabilities and data resources, these tools are poised to play a pivotal role in identifying and prioritizing membrane proteins as personalized anticancer targets.

## 1. Introduction

Cancer remains one of the main causes of death worldwide, being responsible for nearly 10 million deaths each year [1]. There is therefore a continuous need for novel biomarkers and disease-modifying targets. These biomarkers can be leveraged for diagnosis and progression tracking, while the identified targets can be the focus of effective and safe drug development efforts. The etiology of this multifaceted disease often involves aberrant functionality in specific proteins, resulting in increased cellular proliferation and a decrease in standard checkpoints [2]. Notably, membrane proteins have emerged as central players in the development of the most prevalent cancer types [3,4,5]. Unfortunately, their study presents additional challenges compared to their soluble counterparts, as has been extensively reviewed [5,6,7]. On the bright side, the rise of computational methodologies applied to drug discovery in recent decades has provided researchers with a new set of tools to study these protein classes [4,8]. These computational pipelines allow scientists to accelerate and streamline the identification of challenging targets and novel hits by prioritizing experiments and reducing the “wet” experimental burden [9,10].

Computational methods have applications at multiple levels of the oncological drug discovery pipeline [11]. Machine learning (ML) and other statistical models can be used to analyze a wealth of multi-omics data and pinpoint the driver mutations in proteins that may be responsible for the onset of a tumor [12]. These approaches can also be used in the in silico characterization of the effect of point mutations on protein stability, function, and pharmacology [13]. Cancer-related mutations can additionally be analyzed structurally [14], and ML models such as AlphaFold make it possible even when structural data are not available [15]. Furthermore, pharmacophore or quantitative structure–activity relationship (QSAR) models can be used to find chemical structures that either inhibit or activate selectively the target of interest [16,17]. Once a satisfactory selection of candidate molecules is found, structure-based approaches such as molecular docking or molecular dynamics (MD) simulations can help further refine the favorable protein–drug, or in the case of biological drugs, protein–protein interactions [18]. Such detailed knowledge, which many times can only be obtained with computational approaches, is key to enabling personalized oncological treatments [19].

Computational drug discovery pipelines can ease some experimental challenges in membrane protein research, but they face their own issues, mainly stemming from limited data availability due to experimental difficulties [5,8,10]. Different membrane protein families have varying degrees of experimental investigation, particularly in the context of cancer, which together with differences in structural and functional characteristics leads to family-specific challenges. This review will primarily focus on three membrane protein families: receptor tyrosine kinases (RTKs), G protein-coupled receptors (GPCRs), and solute carrier proteins (SLCs), which exhibit differing levels of general understanding and their connection to cancer [20,21,22]. RTKs have been extensively studied, particularly in the context of cancer, with a wealth of research available [23]. GPCRs, on the other hand, have received significant attention in drug discovery, but their connection to cancer has only recently become the subject of investigation [20]. In contrast, SLCs have been relatively understudied in general [24]. These trends explain the amount of literature linking each of these protein families with cancer, computational drug discovery, and both (Figure 1) despite GPCRs and SLCs being the two largest families of membrane proteins.

In the upcoming sections, we first expand on the key experimental and computational challenges particular to the study of membrane proteins. Then, we outline the primary structural and functional characteristics of RTKs, GPCRs, and SLCs, particularly focusing on alterations that are associated with the progression of cancer. Subsequently, we explore the use of multi-omics, ML, and structure-based methods to investigate these anomalies for each protein family, highlighting their inherent challenges. Finally, we place these membrane protein families within the broader landscape of cellular biology, focusing in particular on inter-family crosstalk. To conclude, we delineate potential avenues to further improve the computational characterization of membrane proteins as anticancer targets.

## 2. Key Experimental and Computational Challenges in the Study of Membrane Proteins

Membrane proteins are embedded into the cell membrane yielding an extracellular part, a transmembrane (TM) part, and an intracellular part (Figure 2). It is known that the interactions between the lipids in the cell membrane and the membrane protein are crucial for the protein to acquire the right structure and to function properly [26]. This means that removing the membrane proteins from the membrane lipids, which is necessary for many experimental studies such as protein structural determination, brings additional challenges compared to cytosolic proteins [27,28]. To alleviate a part of this problem, researchers have come up with intricate membrane mimetics that try to imitate the natural environment of the membrane protein instead of using detergents, such as nanodiscs, lipic cubic phase, and styrene malic acid lipid particles [28]. Another problem is the often low expression level of membrane proteins compared to cytosolic proteins. As such, prokaryotes such as Escherichia coli may be used to overexpress a certain membrane protein, but this often leads to aggregation, lack of proper posttranslational modifications, and the misfolding of the membrane protein in the process [27].

Crystallization itself is more difficult for membrane proteins, particularly due to the protein instability outside of the membrane. This means that additional measures are needed, such as the addition of stabilizing molecules, or the construction of protein orthologues, which in turn reduce the throughput of structural characterization [28]. While X-ray crystallization has been the classical method to determine the structure of membrane proteins, cryo-EM has been on the rise, allowing near-native structural determination with resolutions that have vastly increased in recent years [30]. The aforementioned challenges do not equally apply to the protein classes that are reviewed here, as demonstrated by their structural coverage. At the time of writing this review, over 2000 structures were available for 41 out of 58 human RTKs, although in most cases, they only represent the soluble kinase domain [31]. Moreover, 187 out of the 826 GPCRs in the human genome have been structurally determined with a total of 1160 receptor structures in different conformations [32]. However, due to their dynamic nature, under 5% of human SLC protein structures are available—although the number is rapidly increasing thanks to cryo-EM structures [33]. Computational tools, and in particular AlphaFold, are increasingly expanding the availability of good-quality predicted protein structures, although this task is less accurate for protein families with fewer structures available for training, which is the case for membrane proteins [34].

Besides finding the right structure of membrane proteins, it requires additional computing power to be able to simulate the behavior of membrane proteins in their natural environment, with detailed simulations requiring large computing clusters to be achieved [35]. The supplemental amount of interactions that come with the oligomerization of membrane proteins adds to this computing power restraint [36]. Choices have to be made about how the membrane is represented and set up, which lipids are used, and how many atoms are used in the simulation, to name a few issues. Trade-offs have to be made between the amount of detail in the simulation and the time scale at which the simulation takes place [35]. The ever-increasing computing power and efficiency of simulation algorithms help in overcoming these obstacles and, in recent decades, the number of applications has been on the rise, as we review in the following sections.

## 3. Receptor Tyrosine Kinases

RTKs are characterized by a single TM helix, with an extracellular region that recognizes a ligand and an intracellular tyrosine kinase domain (Figure 2a) [22]. This global structure of RTKs is highly evolutionary conserved, both within the human genome and between different species [37]. The main activation of RTKs is through dimerization or oligomerization after binding a ligand. Often, these ligands themselves have a dimeric nature to assist in the activation process [37]. After dimerization, the RTKs are able to auto-phosphorylate each other’s kinase domains. The phosphorylated kinase domain then allows proteins that contain an SH2 domain to bind. The intracellular kinase domain can phosphorylate downstream kinases, which can induce a range of different cellular effects such as cellular differentiation, growth, and proliferation [22]. The aberrant activation of some of these pathways plays a key role in multiple cancers. One example is the Ras/MAPK pathway, which contains the extracellular-signal-regulated ERK5 and p38 kinases downstream. ERK5 is known to play a role in tumor invasion, while p38 is able to regulate the activity of the transcription factor p53, a central protein that is found to be mutated in over half of all tumors [38,39]. Another RTK signaling pathway that often plays a role in cancer is the PI3K/Akt/mTOR pathway. PI3K is phosphorylated by RTKs, after which PI3K phosphorylates Akt. When Akt is activated, it is able to phosphorylate transcription factors such as mTOR which can increase the cell survival and cellular proliferation [38].

The RTK family is the most strongly associated with cancer of the three membrane protein families that are reviewed here. Famous examples of aberrant RTK signaling in cancer are epidermal growth factor receptors HER2 in breast cancer [40] and EGFR in multiple cancers [41,42]. For a more detailed overview of the role of RTKs in cancer, the review by Du et al. is recommended [22]. Many inhibitors of RTKs are currently used in clinics to treat several oncological diseases. As of January 1st, 2024, 43 small molecule RTK inhibitors have been approved for anticancer indications, besides 37 inhibitors targeting other kinase families [43]. Apart from small-molecule drugs, biologics are also used in clinics to combat aberrant RTK signaling. A famous example is the HER2 inhibitor trastuzumab, which is effective in the treatment of HER2+ breast cancer [44].

The main alterations in RTK structure and function leading to cancer development result in increased kinase activity (Figure 3a). Alterations in the kinase domain can increase the stability of the RTK dimer, leading to constitutive activation independent of the ligand [22]. This can be the result of point mutations that are then considered to be drivers of oncogenicity. Indeed, in the case of EGFR, 90% of the mutations in lung cancer are found in the genetic regions that contain the kinase domain [45]. Several mutant driver prediction tools, which rely on different ML tools, are available to forecast the pathogenic effect of these mutations, although their level of agreement is limited. Interestingly, it is higher in RTKs compared to other kinase families, possibly due to the wealth of available training data [46]. Structurally, the effect of mutations in the kinase domain is easier to study, since the intracellular kinase domain of RTKs can be determined experimentally as a soluble protein. For example, structure-based approaches have been able to shed light on the mechanisms of constitutive activation triggered by the D816V mutation in the kinase domain of c-Kit [47]. However, recent studies have highlighted the importance of mutations in the extracellular and TM domains of RTKs [48,49]. Despite the different characteristics of the oncogenic mutations across domains, driver prediction ML models have been able to predict with equal success the oncogenicity of mutations in the extracellular and kinase domains [50]. Moreover, structural studies have allowed us to understand that these mutations in non-kinase domains trigger constitutive ligand-free activation via the covalent extracellular or TM dimerization [50,51].

Constitutive activation can also be the effect of chromosomal rearrangements leading to fusion proteins, which are very common in RTKs. Targeting these fusion proteins in anticancer therapies is very promising because they are not present in healthy cells. Fusion genes can be detected from sequencing data, although the characterization of their oncogenic and druggability potential, and therefore their clinical relevance, is not trivial [52]. Computational analyses of genomic, transcriptomics, and drug sensitivity data can be used to prioritize oncogenic [53] and actionable RTK gene fusions [54]. From validated genetic fusions, ML models can also be constructed to further improve detection, which at the moment still has a problem of high false positive rates. For example, the method developed by Hafstað et al. showed that including an ML-based filter on top of RNA-seq-based fusion detecting algorithms improved the true positive detection rate, although domain-specific information needed to be provided to the model [55]. Deep learning models have also been developed to predict oncogenicity starting from the fusion protein sequence without providing any oncogenic domain-specific features [56].

Given the historical focus on kinase domains, the study of the effect of cancer alterations in substrate affinity on the extracellular domain of the RTK is not very extensive. In fact, there are very few structures of full-length RTKs containing the TM and extracellular domains [57]. However, these can be very useful in designing monoclonal antibodies targeting the ligand-binding region [58]. Structural characterization has also enabled the determination of the mechanism behind the inhibitory synergy of Pertuzumab and Trastuzumab, showing with cryo-EM structures that they do induce cooperative binding [59]. ML models based on structural signatures have been leveraged to improve the design of mAbs [60,61]. The structural analysis of the kinase domain, on the other hand, has been very useful to identify and prioritize small molecules targeting this domain, and to explain the reasons for resistance [62]. The wealth of experimental data for kinase inhibitors has made it possible to use ML models to screen not only potency for wild-type [63] and mutant RTKs [64], but also clinical responses associated with gene expression signatures [65]. Beyond small molecule screening, ML models have also been employed to generate de novo RTK inhibitors by combining 2D and 3D features of known kinase inhibitors [66]. While most drugs are initially designed to bind to a specific target, some drugs bind to multiple kinases, which can be predicted through poly-pharmacology ML models [67]. The combination of ML- and structure-based methods was also leveraged to identify potent small molecules that block the dimerization of the kinase domain [18]. Furthermore, basing these models on structural features has enabled the prediction of drug response toward specific cancer-related RTK mutants [68].

Finally, pathogenic mutations in RTKs can induce the aberrant dimerization or oligomerization that leads to increased signaling. These alterations can be studied through structural analyses. For example, MD simulations helped identify ephrin type-A receptor EphA4 melanoma mutation L920F in the C-terminus as the destabilizing factor leading to receptor trimerization instead of dimerization [69]. Similarly, MD simulations showed that oncogenic mutation V536E in the TM domain of platelet-derived growth factor receptor PDGFRA is responsible for stabilizing a tetrameric conformation responsible for constitutive activation. Further dimerization alterations leading to cancer, such as heterodimerizations of EGFR with other RTKs, have been studied with MD, for which only the kinase domain is needed [70]. Beyond receptor-specific abnormalities, increased RTK signaling in cancer can also be triggered by autocrine and paracrine activation. This is the result of ligand overexpression, which in turn can be studied by multi-omics computational approaches [71].

## 4. G Protein-Coupled Receptors

Ever since the initial characterization of the rhodopsin structure by Schertler et al. [72], a GPCR structure is recognized by its seven TM α-helices, collectively the 7TM domain (Figure 2b). Additionally, it includes an N-terminus, three extracellular loops (ECL), three intracellular loops (ICL), and a C-terminus [73]. Of the three ECLs, ECL2 is usually the longest loop and the most structurally diverse between different GPCRs. An exception is the highly conserved disulfide bond between ECL2 and the TM3 α-helix [74]. Whereas ECL2 is of paramount importance to change the conformation of the GPCR after ligand binding, contacts with the ligand binding to the orthosteric pocket usually happen with ECL1 or ECL3 [74]. The orthosteric binding pocket is on the extracellular side of the 7TM domain and usually comprises TM3, TM5, TM6, and TM7 [75]. Additionally, many allosteric pockets have been described for GPCRs [76]. Upon the activation of the GPCR by a ligand, the structure suffers a conformational rearrangement that primarily involves TM5 and TM6 [75,77]. This conformational change often induces the heterotrimeric G protein that is bound to the ICL2 and ICL3 of the GPCR to exchange its bound GDP for GTP, activating the G protein in the process [75]. Alterations in the GPCR structure due to mutations can lead to, e.g., the constitutive activation of the receptor, where it remains in an active state in the absence of the (endogenous) agonist [74].

GPCRs are the most commonly targeted proteins by drugs, with estimations showing that 35% of all developed drugs have a GPCR as their target [78]. Multiple GPCRs have been extensively associated with cancer, for example, the thyrotropin receptor in thyroid adenomas [79], estrogen receptors GPER1 and GPR30 in breast cancer [80,81], and gonadotropin-releasing hormone receptor GnRH in prostate cancer. In fact, hormonal therapy targeting GnRH is used in the clinic to combat prostate cancer [82]. The smoothened receptor SMO, which is part of the Hedgehog pathway, is also currently targeted by small-molecule antagonists in the treatment of basal cell carcinomas [83]. Furthermore, there is a range of GPCR antagonists that have been or are currently tested in (pre)clinical trials such as Ki16198 LPA receptor inhibitor for pancreatic cancer [84] and astrasentan endothelin receptor inhibitor in prostate cancer [85]. The astrasentan trial, however, like multiple other GPCR antagonists that were developed, had to be ended due to unwanted side-effects occurring in patients [85]. Many clinical candidates target the GPCRs involved in the tumor microenvironment (TME), such as chemokine receptor CXCR4, which are promising targets in immunotherapy [20,83]. For an extensive overview of the current state of GPCR targeting drugs in oncology, the reviews by Arang and Gutkind and Usman et al. are recommended [83,86].

Computational analyses have proven relevant in identifying the role of GPCRs in the TME and their potential role in immunotherapy. For example, the computational analysis of multi-omics data helped pinpoint chemokine receptor axes relevant to particular cancer types and, more importantly, the epigenetic mechanisms responsible for their overexpression (Figure 3b) [87]. Beyond cancer types, GPCR expression signatures extracted with ML models have also been shown to allow head and neck cancer patient stratification into subtypes leading to differential sensitivity to immunotherapy [88]. A similar approach enabled the classification of melanoma patients based on survival and response to immunotherapy based on combined GPCR-TME multi-omics data [89]. These applications have the high potential to define GPCRs as immune biomarkers to help in cancer treatment and patient stratification.

On the tumor side, computational tools can help study the constitutive activation of GPCRs, which may lead to the onset of cancer by inducing downstream cellular pathways [79,90]. An example is the frizzled receptors, which indirectly activate the Wnt pathway, a pathway that is strongly linked to the progression of cancer [91,92]. Genomic analyses have been able to identify oncogenic mutational drivers among GPCR genes, particularly SMO [93]. However, most GPCR mutants do not share the characteristics of classical drivers, such as a high prevalence. With a much lower mutation prevalence than in RTKs, identifying GPCR drivers needed the integration of multi-omics data [94], or the characterization of multi-gene oncodrivers [95], for which computational tools have been crucial. GPCRs of interest in cancer have also been pinpointed solely based on a dysregulated expression in different cancer types, for which there does not seem to be a pan-cancer common profile [96]. Given the challenges of predicting the oncogenic status of GPCR mutations, many authors have opted to study the structural impact of cancer-related mutations on the receptor’s stability and activation mechanism. This can be useful to pinpoint novel potential biomarkers, such as the olfactory receptor OR2T7 destabilizing mutation D125V in glioblastoma [97], or to gain further insights into the activating and binding mechanisms of established anticancer targets, such as CXCR4 [98] or SMO [99], to improve the development of targeted therapies.

The combination of structure-based and ML tools has also made it possible to gain an insight into the different mechanisms behind GPCR involvement in cancer. The principal pathways leading to aberrant GPCR signaling in cancer concern G protein promiscuity and biased signaling [85]. Canonically, every GPCR preferentially activates one of the main four subtypes of G protein α subunits—G_αi_, G_αq_, Gαs, and G_α12/13_. However, some G proteins are more important than others in the development of cancer, which explains why certain cancer-related GPCR mutations lead to G protein promiscuity [85,86]. ML models that predict the probability of different GPCR variants binding to the different Gα protein subunits have been developed, where the GPCR embeddings were generated from the receptor’s sequence [100]. This method enabled the assessment of bias towards different signaling partners beyond G proteins, by also considering β-arrestins as interacting partners, which could have very important implications in the development of biased ligands that preferentially trigger one signaling pathway. However, structurally, there does not seem to be a clear conformation basis for transducer biases [101], which would introduce an important risk of side effects to these therapies.

Similarly to RTKs, the formation of heterodimers has been shown to trigger aberrant GPCR signaling in cancer [102]. The structural analysis of the homo- and heterodimerization patterns and stability therefore introduces novel avenues for treatment. However, in the case of GPCRs, the lipidic environment seems to be extremely determinant in the formation of GPCR oligomers [103], which introduces additional experimental and computational constraints in the mechanistic analyses [104]. Surpassing the technical challenges, however, can help gain insights into cancer-related mutants, leading to distinct di/oligomerization patterns that in turn result in biased signaling [105].

Despite the challenges to computationally assess GPCR oncogenic mechanisms due to the limited availability of training data, the wealth of data collected in non-oncological GPCR drug discovery campaigns is a very good starting point for the discovery of anticancer therapies targeting GPCRs. There are several examples in this area, such as the structure-based virtual screenings of novel small molecules targeting free fatty acid receptor FFAR4 for colorectal cancer [106], or adhesion receptor ADGRF5 for breast cancer [107,108]; or ligand-based screenings of small molecules targeting oxoeicosanoid receptor OXER1 that signal specifically through G_αi_ and/or G_βγ_ for prostate cancer [109]. Beyond providing a wealth of data for novel hit identification, approved GPCR therapies can be considered to be repurposed for oncological applications. To this end, the analysis of omics data can assist in identifying GPCRs with approved drugs that play an important role in cancer survival, such as dopamine receptor 2 in osteosarcoma [110]. Structure-based and ML applications are also common in drug repurposing and can equally be leveraged for cancer applications. However, in the case of GPCRs, there are many risks of off-target effects and unintended implications in the cancer phenotype [111], and thus, omics-aware approaches are preferred.

## 5. Solute Carriers

In contrast to the GPCR and RTK families, which both have recognizable basic structures, the SLC family consists of very diverse proteins. A structure that many SLCs do have in common consists of minimum six but generally 10 to 14 TM α-helices (Figure 2c) [112]. However, due to the difficulties of obtaining SLC protein structures, they are mostly classified on the basis of their known sequence. SLCs are normally considered of the same family when they have an overlap in sequence of at least 20% [112]. ML approaches have been relevant in classifying SLCs into families [113]. While SLC families are structurally diverse, they remain highly evolutionarily conserved within Bilaterian species, with glucose transporters being conserved within all eukaryotes [114]. SLCs do not only have a high sequential and structural variety, but their transport mechanisms are also very diverse, both conformationally and dynamically, which poses a big strain on structure-based methods [24].

SLC transport differing molecules through the cell membrane, such as ions, lipids, and carbohydrates [115]. As Warburg et al. noticed back in 1927, the metabolism in tumor cells differs from that in non-tumor cells [116]. This change in metabolism is achieved through, among other things, changes in the expression of SLCs in the cell [117]. A well-known change in SLC expression in cancer is the upregulation of the glucose transporters to meet the increased demand for glucose in tumor cells [118]. However, no drugs are currently targeting SLCs in an oncological setting. Liu et al. performed a preclinical study in which a glucose transporter GLUT1 inhibitor was able to inhibit cancer cell growth, but this compound was not pursued any further [119]. The monocarboxylate transporter MCT1 inhibitor AZD3965 was tested in a phase 1 clinical trial on patients suffering from advanced stages of cancer, but this drug did not enter phase 2 clinical trials [120].

Multi-omics analyses of SLCs in cancer have been crucial in detecting aberrant mechanisms leading to dysregulated transport rates (Figure 3c). Most commonly, the abnormal expression of SLCs beyond glucose transporters has been associated with the increased transport of metabolites and building blocks necessary for cancer development, which has been used to identify SLCs as prognostic biomarkers in pan-cancer [121,122] and cancer-specific studies [123,124,125]. In this context, ML has also helped further discriminate between the genes with the biggest effect within the cancer signatures [126]. Furthermore, these transcriptomic analyses have been able to identify SLC co-expression patterns that effectively influence cancer development and that can be used as more precise biomarkers than unique SLC signatures [127,128]. Moreover, additional omics data types can help further classify tumor subtypes and make sense of the mechanisms leading to cancer development, for example, by linking the expression profiles to genomic [128,129] or metabolomic data [130,131]. The latter provides an additional advantage since metabolic dysregulation is a good candidate for faster biomarker detection in liquid biopsies [132].

ML and structure-based approaches have also been able to elucidate the role of point mutations triggering changes in transport function, although not frequently in cancer. Polymorphisms in many SLC families are related to several non-oncological diseases, such as cystinuria or ataxia, as well as drug sensitivity, and have mostly been studied in this context [133]. Several analyses have demonstrated the effect of point mutations in SLC structural and functional changes, as well as the potential risk posed by rare uncharacterized mutations [133,134,135,136]. These methods can be further explored in the context of cancer-related mutations, which have also been shown to affect transport efficiency and conformational dynamics [14]. Structural changes can further be exploited to design and virtually screen SLC targeting compounds, as demonstrated for organic anion transporting polypeptides—OATPs [137]. ML-based virtual screening is also possible, but a relative lack of bioactivity data for SLCs is still a big drawback compared to other protein families more broadly explored, such as RTKs and GPCRs [137,138].

## 6. Crosstalk between Membrane Proteins

The fact that proteins do not exist in isolation is one of the most difficult aspects to tackle in cancer research. In turn, crosstalk between proteins can lead to compensation mechanisms, synergistic effects, and therapy resistance. Protein interplay has been extensively characterized for different members of the same family, for example, triggering synergy by co-expression in SLCs [128], or the activation of compensatory networks by RTKs as a mechanism of drug resistance [139]. However, the crosstalk can also happen with members from other membrane protein families (Figure 4), which in turn opens opportunities for novel therapeutic avenues that can further be explored computationally.

Crosstalk between GPCRs and other membrane proteins can lead to oncogenic events. For example, the insulin receptor is known to interact with multiple GPCRs to commence the mTOR pathway [140]. Multiple transactivations between GPCRs and EGFR that induce oncogenic pathways are described, such as GPR30 and EGFR, to activate the oncogenic MAPK and PI3K/Akt pathways or the protease-activated receptor 1 and EGFR in breast cancer [141,142]. Moreover, activated RTKs in cancer have been shown to activate GPCR signaling pathways via direct interaction with G proteins [143]. Computationally, structure-based approaches can be used to gain insights into the mechanisms leading to aberrant signaling [143]. Where RTKs and GPCRs are often concerned with activating downstream proteins to exert an effect, the prime task of SLCs is to transport molecules through the cell membrane, meaning that SLCs themselves cannot activate oncogenic pathways. There are, however, interactions between RTKs and GPCRs with SLCs that aid tumor cells. For example, EGFR is known to be able to stabilize the glucose transporter SGLT1 in tumor cells to increase cell survival [144].

Further crosstalk with SLCs is characterized by shared substrates. This is the case for many GPCR ligands, whereby the expression of SLCs serves as a regulatory mechanism for GPCR ligand availability [145]. An example is monocarboxylate transporter MCT1, which is able to efflux succinate from the cell. This succinate is then able to bind the succinate receptor SUCNR1 (a GPCR), inducing a proinflammatory response [145]. ML models can identify novel and approved small molecules with shared GPCR and SLC targets [146] that can be exploited for drug repurposing or poly-pharmacology approaches in cancer when linked to multi-omics cancer analyses.

Of particular relevance in cancer treatment is the transport of many anti-cancer drugs, including RTK inhibitors such as sunitinib, via SLCs. Thus, alterations in SLCs are a prominent cause of therapy resistance, which can be explored via multi-omics and drug sensitivity analyses [147,148]. Of note, alterations in SLCs—together with other genes—can be responsible not only for resistance to targeted therapies but also for first-line chemotherapy [149]. As a result of these alterations in membrane transporters, not only drug sensitivity is affected, but also prognosis [150], which can help stratify populations for treatment selection.

## 7. Concluding Remarks

Membrane proteins are very promising anticancer targets, but their study is hindered by experimental challenges. Computational tools used in drug discovery pipelines can help overcome some of these challenges, although they are not free of their own obstacles. In particular, the main bottleneck is the lack of experimental data to train ML algorithms or to apply and validate structure-based approaches on. Not all membrane protein families, however, suffer equally from these issues. Historically relevant families such as RTKs have a vast wealth of experimental cancer data and many approved anticancer small molecules, which provide an excellent starting point for ML applications. Moreover, the kinase domain can be experimentally determined and simulated as a soluble kinase, decreasing the threshold for structure-based approaches. This, however, means that the TM and extracellular domains of RTKs are rather unexplored computationally, even though targeting these domains could be key to avoiding off-target effects. Pharmacologically relevant families underexplored in cancer research, such as GPCRs, lack cancer-related data but they compensate for that in non-oncological data. In fact, many computational approaches have been used to study and bring GPCR-targeting molecules to the market. These tools and knowledge can be easily repurposed for oncological applications, although their relevance for this particular applicability domain should be backed up by multi-omics analyses. Finally, in membrane protein families where the lack of experimental data is very prominent, such as SLCs, family-wide tools should be explored that leverage data from other membrane protein families. These can facilitate the prediction of the effect of mutations in TM domains in particular [151,152], or assess the relevance of soluble counterparts of membrane proteins in experimental and computational approaches [153]. Regardless of the wealth of data available for each membrane protein family, they can all benefit from additional computational approaches that consider a holistic view of the tumor and its environment. Some examples include the prediction of the effect of mutations in gene expression [154], or the occurrence of mutant signatures as latent drivers [155], which could be further explored to prioritize personalized cancer therapies [156]. Moreover, the extrapolation of methods beyond their conventional use cases, for example, the application of ML algorithms to analyze structural complexes, can help circumvent some of the classical bottlenecks and assist in the design of novel therapies [157]. In conclusion, computational tools can help analyze the relevance and mechanisms behind membrane protein dysregulation in cancer and will be crucial tools for prioritizing anticancer targets and improved therapies with increasing amounts of data and computational power.

## Figures and Tables

**Figure 1 ijms-25-03698-f001:**
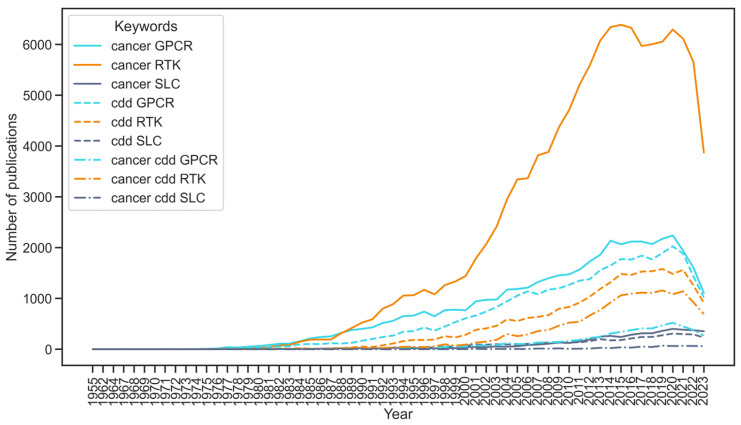
Number of publications in PubMed linking three membrane protein families—Receptor tyrosine kinases (RTKs), G protein-coupled receptors (GPCRs), and solute carriers (SLCs)—to cancer, computational drug discovery (CDD), and the combination of both. The search was computed using the Python package paperscraper [25]. The list of keywords for each term was: cancer—‘cancer’; CDD—‘computational’, ‘computational drug discovery’, ‘artificial intelligence’, ‘deep learning’, ‘machine learning’, ‘expert systems’, ‘QSAR’, ‘PCM’, ‘molecular dynamics’, ‘docking’, ‘molecular modeling’, ‘FEP; RTK—‘RTK’, ‘receptor tyrosine kinase’; GPCR—‘GPCR’, ‘G protein-coupled receptor’; SLC—‘SLC’, ‘solute carrier’.

**Figure 2 ijms-25-03698-f002:**
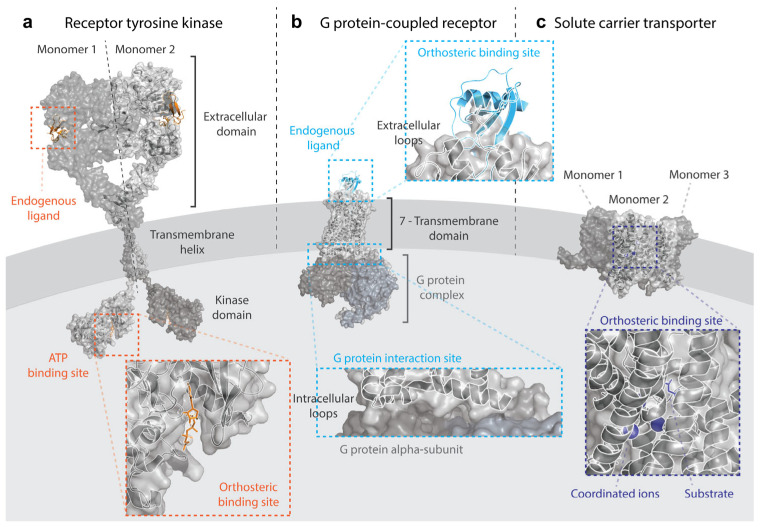
Structural models of three membrane protein family members and their main interacting partners. (**a**) Receptor tyrosine kinases, represented by a dimer model of the epidermal growth factor—EGFR. The model was constructed using the following determined structures from the RCSB Protein Data Bank (PDB): 3NJP for the dimeric extracellular domain in complex with endogenous ligand EFG; 2M20 for the dimeric transmembrane helix; and 2GS6 for the monomeric kinase domain in complex with ATP. (**b**) G protein-coupled receptors, represented by a model of the chemokine receptor CCR2 bound to its endogenous ligand CCL2 and the Gi protein complex. The PDB code used was 7XA3. (**c**) Solute carrier transporters, represented by a trimeric model of the glutamate transporter SLC1A3/EAAT1 in the complex with its endogenous substrate L-Aspartate and in coordination with three sodium ions needed for transport. The PDB code was 7AWM. The models were build using Pymol [29].

**Figure 3 ijms-25-03698-f003:**
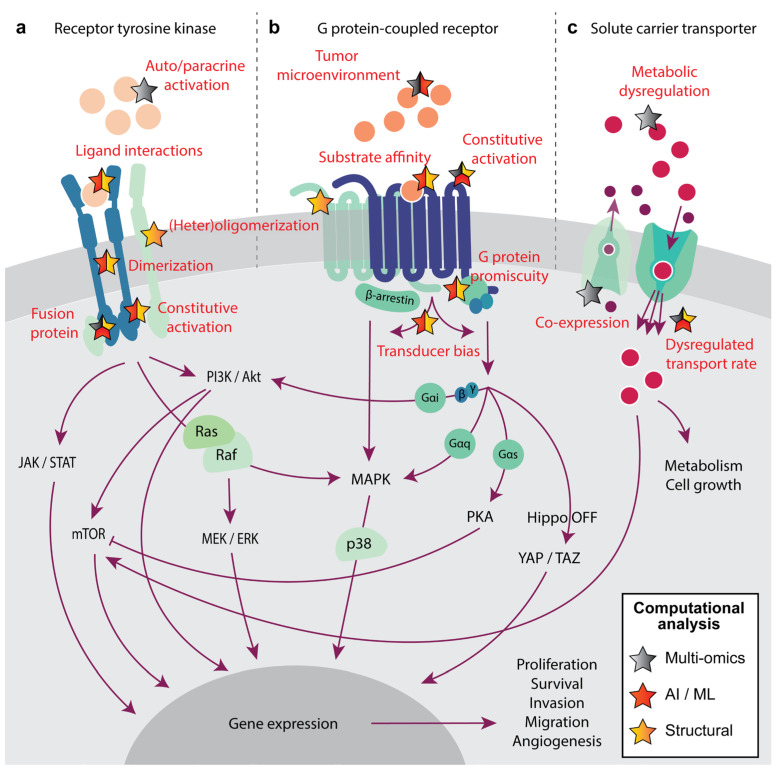
Functional and structural alterations in membrane proteins leading to cancer progression that can be characterized by one or several computational methods, including multi-omics analyses (grey star), artificial intelligence or machine learning algorithms (AI/ML, red star), and structure-based approaches (yellow star). In particular, three membrane protein families are explored: (**a**) Receptor tyrosine kinases—RTK, activated by endogenous ligands represented by light orange spheres; (**b**) G protein-coupled receptors—GPCR activated by endogenous ligands represented by orange spheres, and (**c**) solute carriers—SLC that transport substrates represented by pink and purple spheres.

**Figure 4 ijms-25-03698-f004:**
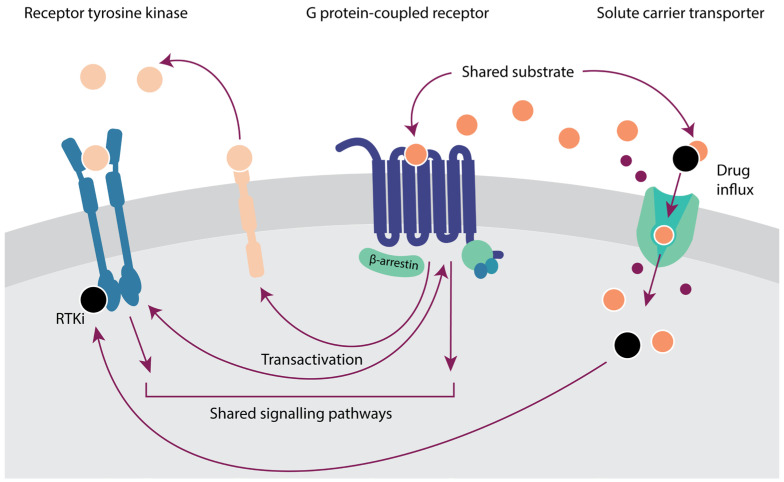
Crosstalk between three membrane protein families—receptor tyrosine kinases (RTKs), G protein-coupled receptors (GPCRs), and solute carriers (SLCs)—in cancer. The expression of RTK endogenous ligands, represented by light orange spheres, can be induced by GPCRs. GPCR ligands, represented by orange spheres, can also be the substrates of SLCs. SLCs also transport RTK inhibitors (RTKi) into the cell.

## Data Availability

The data presented in this study were derived from the following resources available in the public domain: PubMed (https://pubmed.ncbi.nlm.nih.gov/) and RCSB Protein Data Bank (https://www.rcsb.org/).

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
