# Peer review of "Computational Characterization of Membrane Proteins as Anticancer Targets: Current Challenges and Opportunities"

_ijms, 2024, doi:10.3390/ijms25073698_

Round 1

Reviewer 1 Report

Comments and Suggestions for Authors

This manuscript summarizes the characterization of membrane proteins by computational analysis including using AI and/or machine learning which is relatively new techniques. The authors present the new targets suggested by computational analysis clearly and proposed the future directions of this field based on recent papers. The writing is straightforward and it attracted to the readership of the International Journal of Molecular Sciences; however, minor revision of the manuscript is necessary prior to publication.

Comments:
1. Please provide more information of chemical agents (including their chemical structures and properties) which can target the membrane proteins.
2. It would be valuable if the authors could provide the previously reported methods for simulations of protein-protein interactions or protein-ligand interactions by using AI.
3. Please summarize discovered drugs under clinical studies in a table.
4. In conclusions, please discuss how ML can be applied to find a proper chemical structures of potent inhibitors against membrane proteins.
5. Please suggest how to work in both dry-lab and wet-lab ways to develop chemical agents targeting membrane proteins. Moreover, please provide a method to simulate the interactions between membrane proteins and chemicals (or other proteins) more similar to the real reactions.

Author Response

Reviewer 1

This manuscript summarizes the characterization of membrane proteins by computational analysis including using AI and/or machine learning which is relatively new techniques. The authors present the new targets suggested by computational analysis clearly and proposed the future directions of this field based on recent papers. The writing is straightforward and it attracted to the readership of the International Journal of Molecular Sciences; however, minor revision of the manuscript is necessary prior to publication.

We would like to thank the reviewer for taking the time to read and provide feedback on our manuscript. We have carefully considered their suggestions and revised the manuscript accordingly. In general, however, we would like to remind the reviewer that the focus of this review is on the characterization of membrane proteins as anticancer targets, rather than on the development of anticancer drugs targeting membrane proteins. Hereafter, we discuss the modifications made in response to each of the comments provided.

Comments:

  1. Please provide more information of chemical agents (including their chemical structures and properties) which can target the membrane proteins.

Although this would certainly be valuable information in a review where the design of anti-cancer therapies against membrane proteins was discussed, we deemed it outside of the scope of this review. Although in the three protein family sections we consider approved drugs and candidates under development, this is done to provide the reader with background information regarding the druggability of the protein families and the interest of the oncology community in researching these families. Diving deeper into the characteristics of the candidates and approved therapies would shift the focus and hinder readability, therefore we have respectfully not introduced changes in the manuscript in response to this comment.

  1. It would be valuable if the authors could provide the previously reported methods for simulations of protein-protein interactions or protein-ligand interactions by using AI.

This is an interesting point, as we mostly refer in our review to the most conventional uses of AI/machine learning in data-driven applications and structure-based approaches in 3D structural analysis. However, there are indeed many applications of AI-driven methods in structural data. Although the full description of these methods is outside of the scope of this review, we have added a sentence in the concluding remarks to make the reader aware of these methods. Moreover, we have also included a reference to a very recent comprehensive review of ML applications in drug discovery, including structural applications. (Qi et al. - https://doi.org/10.3390/molecules29040903)

  1. Please summarize discovered drugs under clinical studies in a table.

Please, see the response to comment 1.

  1. In conclusions, please discuss how ML can be applied to find a proper chemical structures of potent inhibitors against membrane proteins.

Given the scope of the review, we have restrained from diving too deep into methodologies for therapy design. However, we have taken the opportunity to link this suggestion to comment 2 and mention in the conclusions how ML applications can be relevant not only to characterize the targets but also to assist in the design of novel therapies.

  1. Please suggest how to work in both dry-lab and wet-lab ways to develop chemical agents targeting membrane proteins. Moreover, please provide a method to simulate the interactions between membrane proteins and chemicals (or other proteins) more similar to the real reactions.

Here, we would like to refer back to the scope of this review. Although these are very interesting suggestions, they focus on the rational design of targeted therapies, which is outside the scope of this review and would increase the length of the manuscript significantly.

Reviewer 2 Report

Comments and Suggestions for Authors

In this manuscript van Westen and coworkers present a very nice review article on the advances in the computational characterization of membrane proteins as anticancer targets. Given the characteristic structure and importance for rational design of anticancer drugs of membrane proteins, the authors focused their review on the three families of membrane, i.e., receptor tyrosine kinases (RTKs), G protein-coupled receptors (GPCRs), and solute carrier proteins (SLCs). They summarized and discussed the distinct challenges and opportunities for computational analysis in investigating aberrant protein functionalities associated with cancer progression within each family, in combination of multi-omics data, machine learning, and structure-based methods. This review was well-organized and well written, I would like to recommend publication of the manuscript in its current form in International Journal of Molecular Sciences.   

Author Response

Reviewer 2

In this manuscript van Westen and coworkers present a very nice review article on the advances in the computational characterization of membrane proteins as anticancer targets. Given the characteristic structure and importance for rational design of anticancer drugs of membrane proteins, the authors focused their review on the three families of membrane, i.e., receptor tyrosine kinases (RTKs), G protein-coupled receptors (GPCRs), and solute carrier proteins (SLCs). They summarized and discussed the distinct challenges and opportunities for computational analysis in investigating aberrant protein functionalities associated with cancer progression within each family, in combination of multi-omics data, machine learning, and structure-based methods. This review was well-organized and well written, I would like to recommend publication of the manuscript in its current form in International Journal of Molecular Sciences.  

We would like to thank the reviewer for taking the time to revise our manuscript and for their nice words regarding our work.

Reviewer 3 Report

Comments and Suggestions for Authors

The review paper entitled: "Computational characterization of membrane proteins as anti-cancer targets: Current challenges and opportunities" summarized the literature in regards to the application of several computational techniques on the study of 3 big groups of membrane proteins and their relationship with cancer development and progression. The review is sound and do a good job in giving an overview of the possibilities. My only suggestion would be to include a representative structure of RTKs, GPCRs and SLCs pointing out important regions.

Author Response

Reviewer 3

The review paper entitled: “Computational characterization of membrane proteins as anti-cancer targets: Current challenges and opportunities” summarized the literature in regards to the application of several computational techniques on the study of 3 big groups of membrane proteins and their relationship with cancer development and progression. The review is sound and do a good job in giving an overview of the possibilities. My only suggestion would be to include a representative structure of RTKs, GPCRs and SLCs pointing out important regions.

We would like to thank the reviewer for their time and effort to revise our manuscript. We also appreciate the suggestion to add a figure with the representative structural motifs of the three membrane proteins described. We have added a new Figure 2 to this end.